# Write, Execute, Assess:
# Program Synthesis with a REPL

**Kevin Ellis**[*]
MIT

**Maxwell Nye**[*]
MIT

**Yewen Pu**[*]
MIT

**Felix Sosa**[*]
Harvard University

**Joshua B. Tenenbaum**
MIT

**Armando Solar-Lezama**
MIT

## Abstract

We present a neural program synthesis approach integrating components which write, execute, and assess code to navigate the search space of possible programs. We equip the search process with an interpreter or a read-eval-print-loop (REPL), which immediately executes partially written programs, exposing their semantics. The REPL addresses a basic challenge of program synthesis: tiny changes in syntax can lead to huge changes in semantics. We train a pair of models, a policy that proposes the new piece of code to write, and a value function that assesses the prospects of the code written so-far. At test time we can combine these models with a Sequential Monte Carlo algorithm. We apply our approach to two domains: synthesizing text editing programs and inferring 2D and 3D graphics programs.

## 1 Introduction

When was the last time you typed out a large body of code all at once, and had it work on the first try? If you are like most programmers, this hasn't happened much since "Hello, World." Writing a large body of code is a process of *trial and error* that alternates between trying out a piece of code, executing it to see if you're still on the right track, and trying out something different if the current execution looks buggy. Crucial to this human work-flow is the ability to *execute* the partially-written code, and the ability to *assess* the resulting execution to see if one should continue with the current approach. Thus, if we wish to build machines that automatically write large, complex programs, designing systems with the ability to effectively transition between states of writing, executing, and assessing the code may prove crucial.

In this work, we present a model that integrates components which write, execute, and assess code to perform a stochastic search over the *semantic* space of possible programs. We do this by equipping our model with one of the oldest and most basic tools available to a programmer: an interpreter, or read-eval-print-loop (REPL), which immediately executes partially written programs, exposing their semantics. The REPL addresses a fundamental challenge of program synthesis: tiny changes in syntax can lead to huge changes in semantics. The mapping between syntax and semantics is a difficult relation for a neural network to learn, but comes for free given a REPL. By conditioning the search solely on the execution states rather than the program syntax, the search is performed entirely in the *semantic* space. By allowing the search to branch when it is uncertain, discard branches when they appear less promising, and attempt more promising alternative candidates, the search process emulates the natural iterative coding process used by a human programmer.

In the spirit of systems such as AlphaGo [1], we train a pair of models – a *policy* that proposes new pieces of code to write, and a *value function* that evaluates the long-term prospects of the code

---

[*]These authors contributed equally to this work.

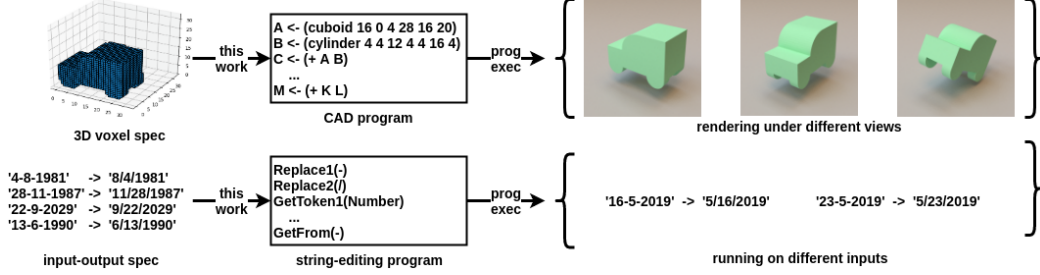

Figure 1: Examples of programs synthesized by our system. Top, graphics program from voxel specification. Bottom, string editing program from input-output specification.

written so far, and deploy both at test time in a symbolic tree search. Specifically, we combine the policy, value, and REPL with a Sequential Monte Carlo (SMC) search strategy at inference time. We sample next actions using our learned policy, execute the partial programs with the REPL, and re-weight the candidates by the value of the resulting partial program state. This algorithm allows us to naturally incorporate writing, executing, and assessing partial programs into our search strategy, while managing a large space of alternative program candidates.

Integrating learning and search to tackle the problem of program synthesis is an old idea experiencing a recent resurgence [2, 3, 4, 5, 6, 7, 8, 9, 10, 11]. Our work builds on recent ideas termed 'execution-guided neural program synthesis,' independently proposed by [12] and [13], where a neural network writes a program conditioned on intermediate execution states. We extend these ideas along two dimensions. First, we cast these different kinds of execution guidance in terms of interaction with a REPL, and use reinforcement learning techniques to train an agent to both interact with a REPL, *and* to assess when it is on the right track. Prior execution-guided neural synthesizers do not learn to *assess* the execution state, which is a prerequisite for sophisticated search algorithms, like those we explore in this work. Second, we investigate several ways of interleaving the policy and value networks during search, finding that an SMC sampler provides an effective foundation for an agent that writes, executes and assesses its code.

We validate our framework on two different domains (see Figure 1): inferring 2D and 3D graphics programs (in the style of computer aided design, or CAD) and synthesizing text-editing programs (in the style of FlashFill [14]). In both cases we show that a code-writing agent equipped with a REPL and a value function to guide the search achieves faster, more reliable program synthesis.

Explicitly, our contributions are:

- We cast the problem of program synthesis as a search problem and solve it using Sequential Monte Carlo (SMC) by employing a pair of learned functions: a value and a policy.

- We equip the model with a REPL, bridging the gap between the syntax and semantics of a program.

- We empirically validate our technique in two different program synthesis domains, text-editing programs and 2D/3D graphics programs, and out-perform alternative approaches.

## 2   An Illustrative Example

To make our framework concrete, consider the following program synthesis task of synthesizing a constructive solid geometry (CSG) representation of a simple 2D scene (see Figure 2). CSG is a shape-modeling language that allows the user to create complex renders by combining simple primitive shapes via boolean operators. The CSG program in our example consists of two boolean combinations: union $+$ and subtraction $-$ and two primitives: circles $C_{x,y}^r$ and rectangles $R_{x,y}^{w,h,\theta}$, specified by position $x, y$, radius $r$, width and height $w, h$, and rotation $\theta$. This CSG language can be formalized as the context-free grammar below:

$$P \;\to\; P \;+\; P \;\mid\; P \;-\; P \;\mid\; C_{x,y}^r \;\mid\; R_{x,y}^{w,h,\theta}$$

The synthesis task is to find a CSG program that renders to *spec*. Our policy constructs this program one piece at a time, conditioned on the set of expressions currently in scope. Starting with an empty

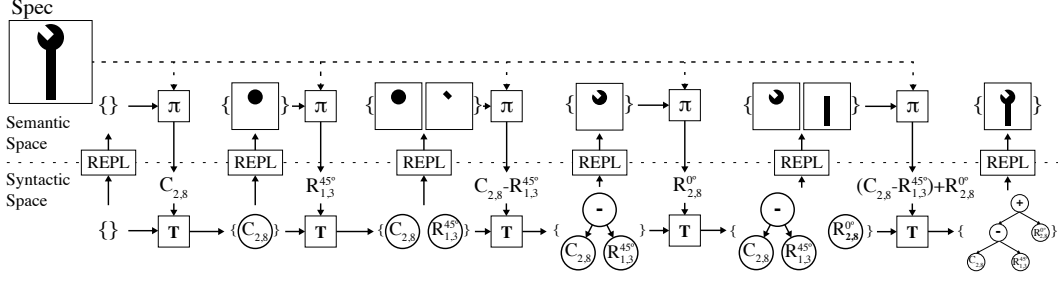

Figure 2: A particular trajectory of the policy building a 2D wrench. At each step, the REPL renders the set of partial programs $pp$ into the semantic (image) space. These images are fed into the policy $\pi$ which proposes how to extend the program via an action $a$, which is incorporated into $pp$ via the transition $T$.

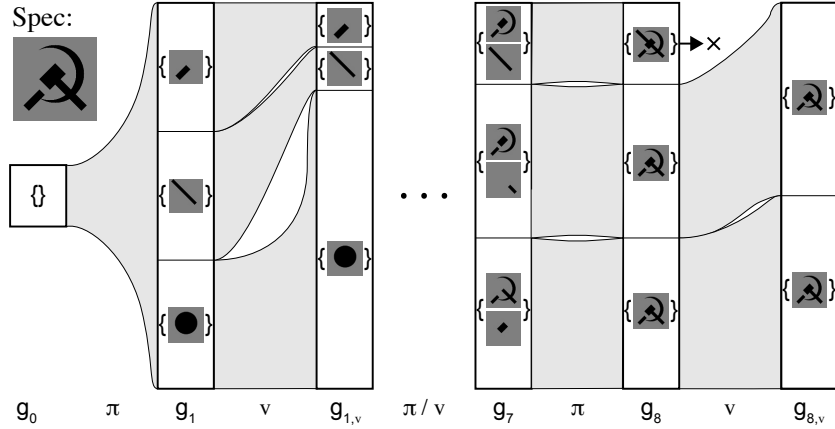

Figure 3: SMC sampler maintains a population of particles (i.e. programs), which it evolves forward in time by (1) sampling from policy $\pi$ to get a new generation of particles, then (2) reweighting by the value function $v$ and resampling to prune unpromising candidates and up-weight promising ones.

*set* of programs in scope, $pp = \{\}$, the policy proposes an action $a$ that extends it. This proposal process is iterated to incrementally extend $pp$ to contain longer and more complex programs. In this CSG example, the action $a$ is either adding a primitive shape, such as a circle $C_{2,8}^{3}$, or applying a boolean combinator, such as $p_1 - p_2$, where the action also specifies its two arguments $p_1$ and $p_2$.

To help the policy make good proposals, we augment it with a REPL, which takes a set of programs $pp$ in scope and executes each of them. In our CSG example, the REPL renders the set of programs $pp$ to a set of images. The policy then takes in the REPL state (a set of images), along with the specification $spec$ to predict the next action $a$. This way, the input to the policy lies entirely in the semantic space, akin to how one would use a REPL to iteratively construct a working code snippet. Figure 2 demonstrates a potential roll-out through a CSG problem using only the policy.

However, code is brittle, and if the policy predicts an incorrect action, the entire program synthesis fails. To combat this brittleness, we use Sequential Monte Carlo (SMC) to search over the space of candidate programs. Crucial to our SMC algorithm is a learned value function $v$ which, given a REPL state, assesses the likelihood of success on this particular search branch. By employing $v$, the search can be judicious about which search branch to prioritize in exploring and withdraw from branches deemed unpromising. Figure 3 demonstrates a fraction of the search space leading up to the successful program and how the value function $v$ helps to prune out unpromising search candidates.

# 3 Our Approach

Our program synthesis approach integrates components that write, execute, and assess code to perform a stochastic search over the semantic space of programs. Crucial to this approach are three main components: First, the definition of the search space; Second, the training of the code-writing policy and the code-assessing value function; Third, the Sequential Monte Carlo algorithm that leverages the policy and the value to conduct the search by maintaining a population of candidate search branches.

## 3.1 The Semantic Search Space of Programs

The space of possible programs is typically defined by a context free grammar (CFG), which specifies the set of syntactically valid programs. However, when one is writing the code, the programs are often constructed in a piece-wise fashion. Thus, it is natural to express the search space of programs as a markov decision process (MDP) over the set of partially constructed programs.

**State**  The state is a tuple $s = (pp, spec)$ where $pp$ is a *set* of partially-constructed program trees (intuitively, 'variables in scope'), and $spec$ is the goal specification. Thus, our MDP is goal conditioned. The start state is $(\{\}, spec)$.

**Action**  The action $a$ is a production rule from the CFG (a line of code typed into the REPL).

**Transitions**  The transition, $T$, takes the set of partial programs $pp$ and applies the action $a$ to either:
1. instantiate a new sub-tree if $a$ is a terminal production: $T(pp, a) = pp \cup \{a\}$
2. combine multiple sub-trees if $a$ is a non-terminal: $T(pp, a) = (pp \cup \{a(t_1 \ldots t_k)\}) - \{t_1 \ldots t_k\}$

Note that in the case of a non-terminal, the children $t_1 \ldots t_k$ are removed, or 'garbage-collected' [13].

**Reward**  The reward is 1 if there is a program $p \in pp$ that satisfies the spec, and 0 otherwise.

Note that the state of our MDP is defined jointly in the syntactic space, $pp$, and the semantic space, $spec$. To bridge this gap, we use a REPL, which evaluates the set of partial programs $pp$ into a semantic or "executed" representation. Let $pp$ be a set of $n$ programs, $pp = \{p_1 \ldots p_n\}$ and let $[\![p]\!]$ denote the execution of a program $p$, then we can write the REPL state as $[\![pp]\!] = \{[\![p_1]\!] \ldots [\![p_n]\!]\}$.

## 3.2 Training the Code-Writing Policy $\pi$ and the Code-Assessing Value $v$

Given the pair of evaluated program states and spec $([\![pp]\!], spec)$, the policy $\pi$ outputs a distribution over actions, written $\pi(a \mid [\![pp]\!], spec)$, and the value function $v$ predicts the expected reward starting from state $([\![pp]\!], spec)$. In our MDP, expected total reward conveniently coincides with the probability of a rollout satisfying $spec$ starting with the partial program $pp$:

$$
\begin{aligned}
v\left([\![pp]\!], spec\right) = \mathbb{E}_\pi\left[R \mid [\![pp]\!], spec\right] &= \mathbb{E}_\pi\left[\mathbb{1}\left[spec \text{ is satisfied}\right] \mid [\![pp]\!], spec\right] \\
&= \mathrm{P}(\text{rollout w/ } \pi \text{ satisfies } spec \mid [\![pp]\!], spec)
\end{aligned}
\tag{1}
$$

Thus the value function simply performs binary classification, predicting the probability that a state will lead to a successful program.

**Pretraining $\pi$.**  Because we assume the existence of a CFG and a REPL, we can generate an infinite stream of training data by sampling random programs from the CFG, executing them to obtain a spec, and then recovering the ground-truth action sequence. Specifically, we draw samples from a distribution over synthetic training data, $\mathcal{D}$, consisting of triples of the spec, the sequence of actions, and the set of partially constructed programs at each step: $(spec, \{a_t\}_{t \leq T}, \{pp_t\}_{t \leq T}) \sim \mathcal{D}$. During pretraining, we maximize the log likelihood of these action sequences under the policy:

$$
\mathcal{L}^{\text{pretrain}}(\pi) = \mathbb{E}_\mathcal{D}\left[\sum_{t \leq T} \log \pi(a_t | [\![pp_t]\!], spec)\right]
\tag{2}
$$

**Training $\pi$ and $v$.**  We fine-tune the policy and train the value function by sampling the policy's roll-outs against $spec \sim \mathcal{D}$ in the style of REINFORCE. Specifically, given $spec \sim \mathcal{D}$, the policy's rollout consists of a sequence of actions $\{a_t\}_{t \leq T}$, a sequence of partial programs $\{pp_t\}_{t \leq T}$, and a

reward $R = \mathbb{1}\left[\llbracket pp_T \rrbracket \text{ satisfies spec}\right]$. Given the specific rollout, we train $v$ and $\pi$ to maximize:

$$\mathcal{L}^{\text{RL}}(v, \pi) = R\sum_{t \leq T} \log v(\llbracket pp_t \rrbracket, spec) + (1 - R)\sum_{t \leq T} \log(1 - v(\llbracket pp_t \rrbracket, spec))$$
$$+ R\sum_{t \leq T} \log \pi(a_t | \llbracket pp_t \rrbracket, spec) \tag{3}$$

### 3.3 An SMC Inference Algorithm That Interleaves Writing, Executing, and Assessing Code

At test time we interleave *code writing*, i.e. drawing actions from the policy, and *code assessing*, i.e. querying the value function (and thus also interleave execution, which always occurs before running these networks). Sequential Monte Carlo methods [15], of which particle filters are the most famous example, are a flexible framework for designing samplers that infer a sequence of $T$ latent variables $\{x_t\}_{t \leq T}$ conditioned on a paired sequence of observed variables $\{y_t\}_{t \leq T}$. Following [4] we construct an SMC sampler for our setting by identifying the policy rollout over time as the sequence of latent variables (i.e. $x_t = pp_t$), and identify the spec as the observed variable at every time step (i.e. $y_t = spec$), which are connected to the policy and value networks by defining

$$P(x_{t+1}|x_t) = \sum_{\substack{a \\ T(x_t,a)=x_{t+1}}} \pi(a|x_t) \qquad\qquad P(y_t|x_t) \propto v(\llbracket x_t \rrbracket, y_t) \tag{4}$$

and, like a particle filter, we approximately sample from $P(\{pp_t\}_{t \leq T}|spec)$ by maintaining a population of $K$ particles – each particle a state in the MDP – and evolve this population of particles forward in time by sampling from $\pi$, importance reweighting by $v$, and then resampling. Unlike MCMC, Sequential Monte Carlo methods do not suffer from 'burn-in' problems, but may need a large population of particles. We repeatedly run our SMC sampler, doubling the particle count each time, until a timeout is reached.

SMC techniques are not the only reasonable approach: one could perform a beam search, seeking to maximize $\log \pi(\{a_t\}_{t<T}|spec) + \log v(\llbracket pp_T \rrbracket, spec)$; or, A* search by interpreting $-\log \pi(\{a_t\}_{t<T}|spec)$ as cost-so-far and $-\log v(\llbracket pp_T \rrbracket, spec)$ as heuristic cost-to-go; or, as popularized by AlphaGo, one could use MCTS jointly with the learned value and policy networks. SMC confers two main benefits: (1) it is a stochastic search procedure, immediately yielding a simple any-time algorithm where we repeatedly run the sampler and keep the best program found so far; and (2) the sampling/resampling steps are easily batched on a GPU, giving high throughput unattainable with serial algorithms like A*.

## 4 Experiments

We study a spectrum of models and test-time inference strategies with the primary goal of answering three questions: Is a value function useful to a REPL-guided program synthesizer; how, in practice, should the value function be used at test time; and how, in practice, does our full model compare to prior approaches to execution-guided synthesis. We answer these questions by measuring, for each model and inference strategy, how efficiently it can search the space of programs, i.e. the best program found as a function of time spent searching.

We test the value function's importance by comparing SMC against policy rollouts[2], and also by comparing a beam search using $\pi$ & $v$ against a beam decoding under just $\pi$; orthogonally, to test how the value function should be best used at test time, we contrast SMC (w/ value) against beam search (w/ value) and also A* (w/ value as the heuristic function). Finally we contrast our best approach—SMC—with recent execution-guided approaches [12, 13], which both perform a beam-search decoding under a learned policy.

As a sanity check, we additionally train a 'no REPL' baseline, which decodes an entire program in one shot using only the spec and intermediate program syntax. This tests whether the REPL helps or hinders. These 'no REPL' architectures follows the prior works CSGNet [16] and RobustFill [8] for CAD and string editing, respectively.

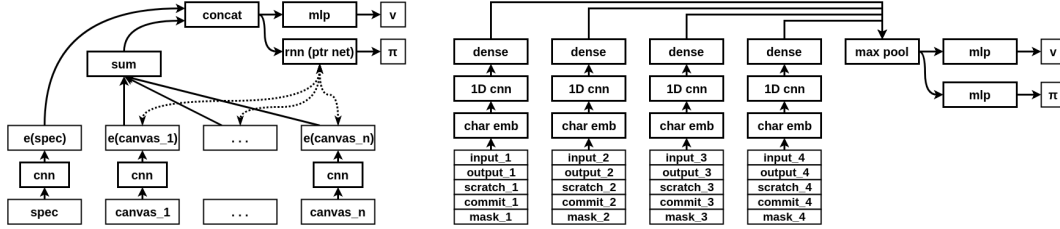

Figure 4: Left: CAD architecture. The policy is a CNN followed by a pointer network [17] (attending over the set of partial programs $pp$) which decodes into the next line of code. The value function is an additional 'head' to the pooled CNN activations. Right: String Editing architecture. We encode each example using an embedding layer, apply a 1-d convolution and a dense block, and pool the example hidden states to predict the policy and value.

## 4.1 Inverse CAD

Modern mechanical parts are created using Computer Aided Design (CAD), a family of programmatic shape-modeling techniques. Here we consider two varieties of *inverse* CAD: inferring programs generating 3D shapes, and programs generating 2D graphics, which can also be seen as a kind of high-level physical object understanding in terms of parts and relations between parts.

We use CSG as our CAD modeling language, where the specification *spec* is an image, i.e. a pixel/voxel array for 2D/3D, and the the goal is to write a program that renders to the target image. These programs build shapes by algebraically combining primitive drawing commands via addition and subtraction, including circles and rotated rectangles (for 2D, w/ coordinates quantized to $16 \times 16$) as well as spheres, cubes, and cylinders (for 3D, w/ coordinates quantized to $8 \times 8 \times 8$), although in both cases the quantization could in principle be made arbitrarily fine. Our REPL renders each partial program $p \in pp$ to a distinct canvas, which the policy and value networks take as input.

**Experimental evaluation** We train our models on randomly generated scenes with up to 13 objects. As a hard test of *out-of-sample generalization* we test on randomly generated scenes with up to 30 or 20 objects for 2D and 3D, respectively. Figure 5 measures the quality of the best program found so far as a function of time, where we measure the quality of a program by the intersection-over-union (IoU) with the spec. Incorporating the value function proves important for both beam search and sampling methods such as SMC. Given a large enough time budget the 'no REPL' baseline is competitive with our ablated alternatives: inference time is dominated by CNN evaluations, which occur at every step with a REPL, but only once without it. Qualitatively, an integrated policy, value network, and REPL yield programs closely matching the spec (Figure 6). Together these components allow us to infer very long programs, despite a branching factor of $\approx$1.3 million per line of code: the largest programs we successfully infer[3] go up to 21 lines of code/104 tokens for 3D and 29 lines/158 tokens for 2D, but the best-performing ablations fail to scale beyond 11 lines/59 tokens for 3D and 19 lines/117 tokens for 2D.

## 4.2 String Editing Programs

Learning programs that transform text is a classic program synthesis task [18] made famous by the FlashFill system, which ships in Microsoft Excel [14]. We apply our framework to string editing programs using the RobustFill programming language [8], which was designed for neural program synthesizers.

**REPL** Our formulation suggests modifications to the RobustFill language so that partial programs can be evaluated into a semantically coherent state (i.e. they execute and output something meaningful). Along with edits to the original language, we designed and implemented a REPL for this domain, which, in addition to the original inputs and outputs, includes three additional features as part of the intermediate program state: The **committed** string maintains, for each example input, the output of the expressions synthesized so far. The **scratch** string maintains, for each example input, the partial results of the expression currently being synthesized until it is complete and ready to be

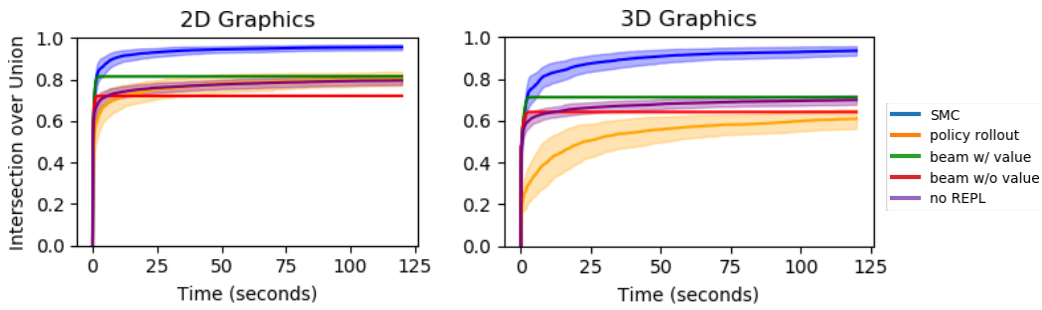

Figure 5: Quantitative results for CAD on out-of-sample testing problems. Both models trained on scenes with up to 13 objects. Left: 2D models tested on scenes with up to 30 objects. Right: 3D models tested on scenes with up to 20 objects. A* unapplicable due to extremely large action space (here, branching factor > 1.3 million). Error bars: average stddev for sampling-based inference procedures over 5 random seeds. SMC achieves the highest test accuracy.

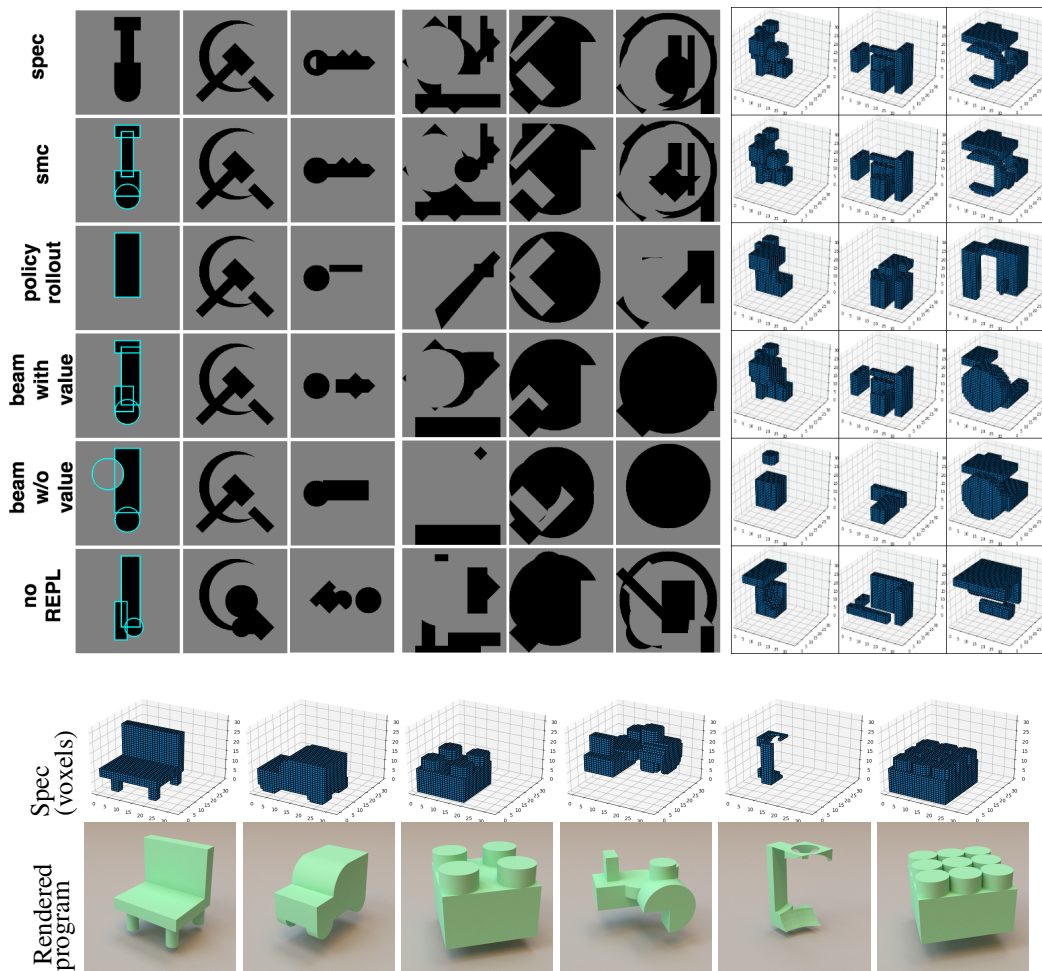

Figure 6: Qualitative inverse CAD results. Top: Derendering random scenes vs. ablations and no-REPL baseline. Teal outlines show shape primitives in synthesized 2D programs. Bottom: Rerendering program inferred from voxels from novel viewpoints.

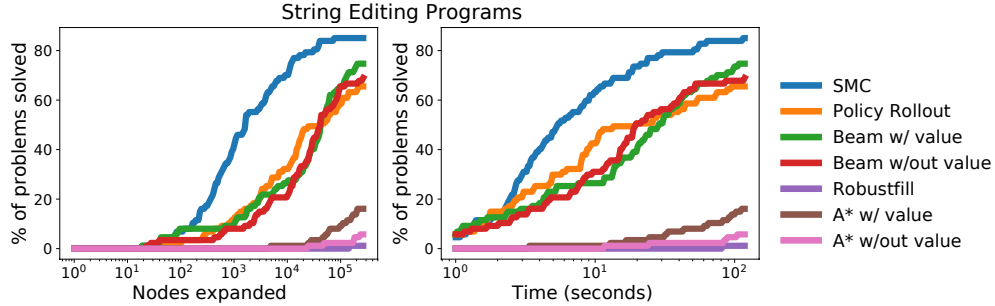

Figure 7: Results for String Editing tasks. Left: tasks solved vs number of nodes expanded. Right: tasks solved vs total time per task. Our SMC-based search algorithm solves more tasks using 10x fewer node expansions and less time than previous approaches. Note that x-axes are log scale.

| Spec: | | Spec: | |
| --- | --- | --- | --- |
| 6/12/2003 | → date: 12 mo: 6 year: 2003 | Dr Mary Jane Lennon → Lennon, Mary Jane (Dr) | |
| 3/16/1997 | → date: 16 mo: 3 year: 1997 | Mrs Isaac McCormick → McCormick, Isaac (Mrs) | |
| Held out test instance: | | Held out test instance: | |
| 12/8/2019 | → date: 8 mo: 12 year: 2019 | Dr James Watson → Watson, James (Dr) | |
| Results: | | Results: | |
| **SMC (Ours)** | **→ date: 8 mo: 12 year: 2019** | **→ Watson, James (Dr)** | |
| Rollout | → date: 8 mo: 1282019 | → Watson, James | |
| Beam w/value | → date: 8 mo: 12 year:2019 | → Watson, JamesDr | |
| Beam | → date: 8 mo: 12 year: | → Watson, James ( | |
| RobustFill | → date:12/8/2019 | → Dr James Watson | |

Figure 8: Comparison of best programs on held-out inputs. Best programs determined by Levenshtein distance of program outputs to spec outputs. Leveraging the policy network, value network, and REPL-based execution guidance, SMC is able to consistently find programs with the desired behavior.

added to the *committed* string. Finally, the binary valued **mask** features indicate, for each character position, the possible locations on which transformations can occur.

**Experimental Evaluation**  We trained our model and a reimplementation of RobustFill on string editing programs randomly sampled from the CFG. We originally tested on string editing programs from [6] (comprising training tasks from [4] and the test corpus from [19]), but found performance was near ceiling for our model. We designed a more difficult dataset of 87 string editing problems from 34 templates comprising address, date/time, name, and movie review transformations. This dataset required synthesis of long and complex programs, making it harder for pure neural approaches such as RobustFill.

The performance of our model and baselines is plotted in Figure 7, and examples of best performing programs are shown in Figure 8. The value-guided SMC sampler leads to the highest overall number of correct programs, requiring less time and fewer nodes expanded compared to other inference techniques. We also observe that beam search attains higher overall performance with the value function than beam search without value. Our model demonstrates strong out-of sample generalization: Although it was trained on programs whose maximum length was 30 actions and average length approximately 8 actions, during test time we regularly achieved programs with 40 actions or more, representing a recovery of programs with description length greater than 350 bits.

## 5  Discussion

**Related Work**  Within the program synthesis community, both text processing and graphics program synthesis have received considerable attention [14].We are motivated by works such as InverseCSG [20], CSGNet [16], and RobustFill [8], but our goal is not to solve a specific synthesis problem in isolation, but rather to push toward more general frameworks that demonstrate robustness across domains.

We draw inspiration from recent neural "execution-guided synthesis" approaches [13, 12] which leverage partial evaluations of programs, similar to our use of a REPL. We build on this line of work by explicitly formalizing the task as an MDP, which exposes a range of techniques from the RL and planning literatures. Our addition of a stochastic sampling inference procedure and a learned value network demonstrates marked improvements on methods that do not leverage these techniques. Prior work [21] combines tree search with $Q$-learning to synthesize small assembly programs, but do not scale to large programs with extremely high branching factors, as we do (e.g., the $> 40$ action-programs we synthesize for text editing or the $>1.3$ million branching factor per line of code in our 3D inverse CAD).

We train our models and baselines with only the simplest reinforcement learning techniques, but the literature contains many sophisticated techniques for jointly learning policy and value functions, such as asynchronous actor-critic methods [22]. While our simple approach suffices to answer the questions our work poses, namely whether and how a REPL and policy/value networks are useful, we anticipate that scaling up our approach could benefit from these recent modern advances in deep reinforcement learning.

**Outlook** We have presented a framework for learning to write code combining two ideas: allowing the agent to explore a tree of possible solutions, and executing and assessing code as it gets written. This has largely been inspired by previous work on execution-guided program synthesis, value-guided tree search, and general behavior observed in how people write code.

An immediate future direction is to investigate programs with control flow like conditionals and loops. A Forth-style stack-based [23] language offer promising REPL-like representations of these control flow operators. But more broadly, we are optimistic that many tools used by human programmers, like debuggers and profilers, can be reinterpreted and repurposed as modules of a program synthesis system. By integrating these tools into program synthesis systems, we believe we can design systems that write code more robustly and rapidly like people.

**Acknowledgments**

We gratefully acknowledge many extended and productive conversations with Tao Du, Wojciech Matusik, and Siemens research. In addition to these conversations, Tao Du assisted by providing 3D ray tracing code, which we used when rerendering 3D programs. Work was supported by Siemens research, AFOSR award FA9550-16-1-0012 and the MIT-IBM Watson AI Lab. K. E. and M. N. are additionally supported by NSF graduate fellowships.

## Footnotes

[2]SMC w/o a value function would draw samples from $\pi$, thus the relevant comparison for measuring the value of the value function is SMC w/ value against policy rollouts.

[3]By "successfully infer" we mean IoU$\geq$0.9

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
