[Supplementary Material]

# A Appendix

## A.1 Graphics Programming Language

**CFG** The programs are generated by the CFG below:

```
P  −>  P + P  |  P − P  |  S

# For 2D graphics
S  −>  circle(radius=N, x=N, y=N)
   |  quadrilateral(x0=N, y0=N,
                    x1=N, y1=N,
                    x2=N, y2=N,
                    x3=N, y3=N)

N  −>  [0 : 31 : 2]

# For 3D graphics
S  −>  sphere(radius=N, x=N, y=N, z=N)
   |  cube(x0=N, y0=N, z0=N,
           x1=N, y1=N, z1=N)
   |  cylinder(x0=N, y0=N, z0=N,
               x1=N, y1=N, z1=N, radius=N)
N  −>  [0 : 31 : 4]
```

In principle the 2D language admits arbitrary quadrilaterals. When generating synthetic training data we constrain the quadrilaterals to be take the form of rectangles rotated by 45 increments, although in principle one could permit arbitrary rotations by simply training a higher capacity network on more examples.

## A.2 String Editing Programming Language

**CFG** Our modified string editing language, based on [8] is defined as follows:

```
Program P  −>  concat(E,.., E)  (at most 6 E's)
Expression E  −>  F | N | N(N)
              |  N(F) | Const
Substring F  −>  Sub1(k) Sub2(k)
              |  Span1(r) Span2(i) Span3(y)
                 Span4(r) Span5(i) Span6(y)
Nesting N  −>  GetToken1(t) GetToken2(i)
           |  ToCase(s) | GetUpTo(r)
           |  GetFrom(r) | GetAll(t)
           |  GetFirst1(t) GetFirst2(i)
Regex r  −>  t | d
Type t  −>  Number | Word | AlphaNum
         |  Digit | Char | AllCaps
         |  Proper | Lower
Case s  −>  AllCaps | PropCase | Lower
Delimiter d  −>  &,.?!@()[]%{}/#$:;"'
Index i  −>  {−5 .. 6}
Boundary y  −>  Start | End
```

## A.3 Training details

**String Editing** We performed supervised pretraining for 24000 iterations with a batch size of 4000. We then performed REINFORCE for 12000 epochs with a batch size of 2000. Training took approximately two days with one p100 GPU. We use the Adam optimizer wiht default settings.

Our Robustfill baseline was a re-implementation of the "Attn-A" model from [8]. We implemented the "DP-beam" feature, wherein during test-time beam search, partial programs which lead to an

output string which is not a prefix of the desired output string are removed from the beam. We trained for 50000 iterations with a batch size of 32. Training also took approximately two days with one p100 GPU.

**2D/3D Graphics**   We performed supervised pretraining with a batch size of 32, training on a random stream of CSG programs with up to 13 shapes, for approximately three days with one p100 GPU. We use the Adam optimizer wiht default settings. Over three days the 3D model saw approximately 1.7 million examples and the 2D model saw approximately 5 million examples. We fine-tuned the policy using REINFORCE and trained the value network for approximately 5 days on one p100 GPU. For each gradient step during this process we sampled $B_1 = 2$ random programs and performed $B_2 = 16$ rollouts for a total batch size of $B = B_1 \times B_2 = 32$. During reinforcement learning the 3D model saw approximately 0.25 million examples and the 2D model saw approximately 9 million examples.

For both domains, we performed no hyperparameter search. We expect that with some tuning, results could be marginally improved, but our goal is to design a general approach which is not sensitive to fine architectural details.

## A.4   Data and test-time details

For both domains, we used a 2-minute timeout for testing for each problem, and repeatedly doubled the beam/number of particles until timeout is reached.

**String Editing**   We originally tested on string editing programs from [6] (comprising training tasks from [4] and the test corpus from [19]), but found our performance was near ceiling for our model (Figure 10). Thus, we designed our own dataset, as described in the main text. Generation code for this dataset can be found in our supplement, in the file `generate_test_robust.py`.

**2D/3D Graphics**   We generate a scene with up to $k$ objects by sampling a number between 1 to $k$, and then sampling a random CSG tree with that many objects. We then remove any subtrees that do not affect the final render (e.g., subtracting pixels from empty space). Our held-out test set is created by sampling 30 random scenes with up to $k = 20$ objects for 3D and $k = 30$ objects for 2D. Running `python driver.py demo -maxShapes 30` using the attached supplemental source code will generate example random scenes. Figure 9 illustrates ten random 3-D/2-D scenes and contrasts different model outputs.

## A.5   Architecture details

### A.5.1   String Editing

For this domain, our neural architecture involves encoding each example state separately and pooling into a hidden state, which is used to decode the next action. To encode each example, we learn an embedding vector of size 20 for each character and apply it to each position in the input string, output string, committed string, and scratch string. For each character position, we concatenate these embedding vectors, additionally concatenating the values of the masks for that spatial position. We then perform a 1-d convolution with kernel size 5 across the character positions. Following [13], we concatenate the vectors for all the character positions, and pass this through a dense block with 10 layers and a growth rate of 128 to produce a hidden vector for a single example. We perform an average pooling on the hidden vector for each example. We then concatenate the resulting vector with a 32-dim embedding of the previous action and apply a linear layer, which results in the final state embedding, from which we decode the next action. Our value network is identical, except the final layer instead decodes a value.

### A.5.2   Inverse CAD

The policy is a CNN followed by a pointer network which decodes into the next line of code. A pointer network [17] is an RNN that uses a differentiable attention mechanism to emit pointers, or indices, into a set of objects. Here the pointers index into the set of partial programs $pp$ in the current state, which is necessary for the union and difference operators. Because the CNN takes as input the current REPL state – which may have a variable number of objects in scope – we encode each object

with a separate CNN and sum their activations, i.e. a 'Deep Set' encoding [24]. The value function is an additional 'head' to the pooled CNN activations.

Concretely the neural architecture has a *spec encoder*, which is a CNN inputting a single image, as well as a *canvas encoder*, which is a CNN inputting a single canvas in the REPL state, alongside the spec, as a two-channel image. The canvas encoder output activations are summed and concatenated with the spec encoder output activations to give an embedding of the state:

$$\text{stateEncoding}(spec, pp) = \text{specEncoder}(spec) \otimes \sum_{p \in pp} \text{canvasEncoder}(\text{spec}, [\![\text{p}]\!]) \tag{5}$$

for $W_1$, $W_2$ weight matrices.

For the policy we pass the state encoding to a pointer network, implemented using a GRU with 512 hidden units and one layer, which predicts the next line of code. To attend to canvases $p \in pp$, we use the output of the canvas encoder as the 'key' for the attention mechanism.

For the value function we passed the state in coding to a MLP with 512 hidden units w/ a hidden ReLU activation, and finally apply a negated 'soft plus' activation to the output to ensure that the logits output by the value network is nonpositive:

$$v(spec, pp) = -\text{SoftPlus}(W_2\text{ReLU}(W_1\text{stateEncoding}(spec, pp))) \tag{6}$$

$$\text{SoftPlus}(x) = \log(1 + e^x) \tag{7}$$

**2-D CNN architecture:** The 2D spec encoder and canvas encoder both take as input $64 \times 64$ images, passed through 4 layers of 3x3 convolution, with ReLU activations after each layer and 2x2 max pooling after the first two layers. The first 3 layers have 32 hidden channels and the output layer has 16 output channels.

**3-D CNN architecture:** The 3D spec encoder and canvas encoder both take as input $32 \times 32 \times 32$ voxel arrays, passed through 3 layers of 3x3 convolution, with ReLU activations after each layer and 4x4 max pooling after the first layer. The first 2 layers have 32 hidden channels and the output layer has 16 output channels.

**No REPL baseline:** Our "No REPL" baselines using the same CNN architecture to take as input the *spec*, and then use the same pointer network architecture to decode into the program, with the sole difference that, rather than attend over the CNN encodings of the objects in scope (which are hidden from this baseline), the pointer network attends over the hidden states produced at the time when each previously constructed object was brought into scope.

## A.6  String editing additional results

Figure 10 shows results on the string editing dataset from [6].

Figure 9: Top: Spec. Bottom rows (in order): SMC, policy rollouts, beam search with value, beam search without value, no REPL

Figure 10: Results for String Editing tasks on dataset from [6]. Left: tasks solved vs number of nodes expanded. Right: tasks solved vs total time per task. Note that x-axes are log scale.

inputs:
('3/16/1997', '4/17/1986', '6/12/2003', '4/23/1997')
outputs:
('date: 16 mo: 3 year: 1997', 'date: 17 mo: 4 year: 1986', 'date: 12 mo: 6 year: 2003', 'date: 23 mo: 4 year: 1997')
Const(d), Commit, Const(a), Commit, Const(t), Commit, Const(e), Commit, Const(:), Commit, Const( ), Commit, Replace1(/), Replace2( ), GetToken1(Number), GetToken2(1),
Commit, Const( ), Commit, Const(m), Commit, Const(o), Commit, Const(:), Commit, Const( ), Commit, GetUpTo(Number), Commit, Const( ), Commit, Const(y), Commit,
Const(e), Commit, Const(a), Commit, Const(r), Commit, Const(:), Commit, Const( ), Commit, GetFrom(/), Commit

inputs:
('April 19, 2:45 PM', 'July 5, 8:42 PM', 'July 13, 3:35 PM', 'May 24, 10:22 PM')
outputs:
('April 19, approx. 2 PM', 'July 5, approx. 8 PM', 'July 13, approx. 3 PM', 'May 24, approx. 10 PM')
GetUpTo( ), Commit, GetFirst1(Number), GetFirst2(-3), Commit, Const(,), Commit, Const( ), Commit, Const(a), Commit, Const(p), Commit, Const(p), Commit, Const(r),
Commit, Const(o), Commit, Const(x), Commit, Const(.), Commit, Const( ), Commit, GetFrom(,), GetFirst1(Digit), GetFirst2(3), GetFirst1(Digit), GetFirst2(-3), Commit,
Const( ), Commit, Const(P), Commit, Const(M), Commit

inputs:
('cell: 322-594-9310', 'home: 190-776-2770', 'home: 224-078-7398', 'cell: 125-961-0607')
outputs:
('(322) 5949310 (cell)', '(190) 7762770 (home)', '(224) 0787398 (home)', '(125) 9610607 (cell)')
Const((), Commit, ToCase(Proper), GetFirst1(Number), GetFirst2(1), GetFirst1(Char), GetFirst2(2), Commit, Const()), Commit, Const( ), Commit, GetFirst1(Number),
GetFirst2(5), GetFirst1(Char), GetFirst2(-2), GetToken1(Char), GetToken2(3), Commit, SubStr1(-16), SubStr2(17), GetFirst1(Number), GetFirst2(4), GetToken1(Char),
GetToken2(-5), Commit, GetFirst1(Number), GetFirst2(5), GetToken1(Char), GetToken2(-5), Commit, GetToken1(Number), GetToken2(2), Commit, Const( ), Commit,
Const((), Commit, GetUpTo(-), GetUpTo(Word), Commit, Const()), Commit

inputs:
('(137) 544 1718', '(582) 431 0370', '(010) 738 6792', '(389) 820 9649')
outputs
('area code: 137, num: 5441718', 'area code: 582, num: 4310370', 'area code: 010, num: 7386792', 'area code: 389, num: 8209649')
Const(a), Commit, Const(r), Commit, Const(e), Commit, Const(a), Commit, Const( ), Commit, Const(c), Commit, Const(o), Commit, Const(d), Commit, Const(e), Commit,
Const(:), Commit, Const( ), Commit, GetFirst1(Number), GetFirst2(0), Commit, Const(,), Commit, Const( ), Commit, Const(n), Commit, Const(u), Commit, Const(m),
Commit, Const(:), Commit, Const( ), Commit, GetFrom()), GetFirst1(Number), GetFirst2(2), Commit

Figure 11: Examples of long programs inferred by our system in the string editing domain.