[Reviews · NeurIPS 2019]

Reviewer 1



This line of work is very promising, but it seems like a small variation from existing work in the area. As an example, how does this paper compare to [Tian et al. 2019] that has a very similar approach and similarly examines 3d graphics? As far as the method itself. One area of confusion is in the reward signal. Page 3 line 129 claims that the reward signal is 1 if the partial program satisfies the spec. Clearly that cannot be entirely correct, because it must give some reward to partial solutions. Indeed, Page 6 line 177 states that IoU is the measure used. It would be useful to be more clear exactly how this works.

Reviewer 2



> "Given a large enough time budget the ‘no REPL’ baseline is competitive with our ablated alternatives." - As per my understanding, the key result here is the effectiveness of search (SMC) over pure RL methods in the program synthesis context. However, the policy rollout baseline is trained with RL using a single machine, making it difficult to explore using entropy based methods or epsilon greedy. However, using multiple actors in an asynchronous setting would be a stronger/fairer baseline (and then doing policy rollouts) to the SMC approach. I expect SMC to do well but this is an important empirical question (other methods cited like Ganin et al. seem to do this in the same context). > "The value-guided SMC sampler leads to the highest overall number of correct programs, requiring less time and fewer nodes expanded compared to other inference techniques. " -> how well does a SMC sampler work without value guided proposals for both case studies? > How sensitive are results in Figure 6 to random seeds? > Are there burn-in issues to get the sampling procedure to work reliably? > It will be informative to visualize and show the SMC population for the 3D examples, similar to Figure 3

Reviewer 3



The proposed idea of partial program evaluation based on a REPL seems to be based on a careful design of the evaluation mechanism and is specific to each of the two domains targeted. Even for the domains considered in the paper, like the graphics domain I think different evaluation mechanisms may have different effects on the synthesis outcomes. The proposed way of generation of a program starts from the bottom of the syntax tree and generates the terminals (individual shapes), which can be evaluated quite well, but if we follow the grammar and synthesize the program top-down from the non-terminals then the partial evaluation is not as easy. I feel some discussion on the effect of different possible evaluation mechanisms and different designs of the sequential decision process can be helpful. The RL loss in eq (3) is somewhat non-standard, I wonder if the authors have tried more standard value estimation losses rather than the cross-entropy loss. The paper is well written and clear.

[Author Response · NeurIPS 2019]

We thank our reviewers for taking the time to critique and improve our paper.

Reviewer 1 (R1) suggests comparing with related 3D graphics synthesis work, particularly Tian et al. 2019. Although our submission cites Tian et al., we agree that this work deserves a more extended discussion. Like us, they introduce an approach to inferring programmatic 3D models of objects. A key similarity between our works is that we both condition on intermediate renders, a feature also shared with other works in the computer vision literature such as Eslami et al. 2016 ("attend-infer-repeat"); Richtie et al. 2016; Ellis et al. 2018; and Ganin et al. 2018 ("SPIRAL"). Our revision will discuss these similarities. However, we disagree that our approach is a "small variation" of this work, as R1 claims. In particular, Tian et al. takes a sophisticated approach that is nonetheless *specialized* to graphics programs, while we propose a *general* framework for program synthesis, allowing us to also solve very different problems such as FlashFill-style text manipulation. Specifically, the success of Tian et al. hinges on exploiting the differentiability of the rendering process to train a program generating policy and generalize it beyond the training distribution via fine-tuning by gradient descent. In contrast, we assume a generic REPL that is not differentiable, and generalize beyond the training distribution by leveraging search (SMC) at test time. Furthermore, the goals and problem statement of our 3D work differ enough from those of Tian et al. to preclude a head-to-head experimental comparison. Our goal is generic geometric shape reconstruction from low-level primitives via algebraic manipulations (i.e. union/difference). In Tian et al., the goal is to decompose everyday objects into sub-parts and symmetries; all components are 'unioned' together, and they assume specialized semantic primitives such as 'Leg' and 'ChairBeam'.

Instead, we feel the most appropriate related work to experimentally compare with is CSGNet (Sharma et al. 2018) for graphics programs and RobustFill (Devlin et al. 2017) for text editing, because they solve the same kinds of inverse CSG and text editing problems we do. We compared with our reimplementation of CSGNet (Figure 5 of our submission) and with a reimplementation of RobustFill (Figure 7 of our submission), in both cases finding that a REPL with learned stochastic search outperforms these alternatives. Our text editing test cases are especially challenging for RobustFill: we originally evaluated on existing text editing benchmarks from the literature (Nye et al. 2019), but found that our REPL approach was at ceiling (98% solved) for these benchmarks (line 204 of submission & Fig. 10 of appendix), and so needed to design an even more challenging benchmark suite (Fig. 7-8 & Appendix Fig. 11). For these reasons, we believe that our current baseline results adequately answer the questions our paper poses, namely whether a REPL is useful to a code-writing agent, and how to integrate the REPL at test time with a symbolic search procedure.

R1 points out a spot of possible confusion: during training, we use a 0/1 reward function without partial credit, but during testing, we use graded metrics (intersection-over-union for inverse CAD) to evaluate the performance of baselines. We are able to successfully use a 0/1 reward during training because we bootstrap our policy with imitation learning.

Reviewer 2 (R2) suggests trying more sophisticated RL training procedures. For instance, one could use Actor-Critic methods to train $\pi$ and $v$, *both* for our full model, *and* for our baselines and ablations; or, one could use SMC *at train time*, like how AlphaGoZero uses MCTS both at train and test. We opted for vanilla REINFORCE as the simplest algorithm to answer the question of how to employ a REPL with learned search, and we will discuss these next steps in a revision. Learning the policy $\pi$ without imitation would be difficult due to the large action space ($> 1.3$ million different actions for 3D CAD, line 183 of paper), which results in very sparse reward signals even with search.

R2 asks several questions about our SMC sampler that will be clarified in the revision. SMC without value guided proposals would be equivalent to drawing samples from the policy; unlike MCMC methods, SMC does not suffer from "burn-in" or "mixing" issues, but may need a large population of particles (we repeatedly double the particle count until timeout); the SMC seed slightly affects the time to recover the results in Figure 6 (see graph to right: error bars over 5 random seeds and correspond to stddev), but typically leaves the final rerendering unaffected.

Reviewer 3 (R3) asks good questions about the order of syntax tree generation, particularly regarding the string editing domain. We chose bottom-up generation because it allows execution of partially written programs, and most closely mirrors how people write code in a REPL. For string editing, we kept the semantics of the RobustFill DSL the same but modified the syntax (Appendix A.2) to ensure that intermediate executions are available during bottom-up generation. To compare to other forms of program generation, a recent piece of related work, (Nye et al. 2019) considers *top-down* generation of programs, in which partial programs cannot be executed. As noted above, our performance on the dataset of Nye et al. far exceeds the performance of their top-down system (98% solved vs. 77% solved), which we believe helps show the value of a REPL-based approach that can execute code on-the-fly as it is written.

R3 points out that our value estimation loss is nonstandard. This is because our expected future reward coincides directly with the probability of the agent generating a correct program (discussed in lines 116-119 of paper). Thus, it is natural to employ a cross-entropy loss because we are predicting a probability: unlike expected reward in e.g. Atari games such as Pong, our expected reward has a probabilistic interpretation.

[Meta-Review · NeurIPS 2019]

This paper provides a method for Deep RL-based program synthesis, which exploits a SMC sampler during inference progressively decode into an executable program. The reviewers were enthusiastic about this method and found the experimental support for the proposal convincing. Without much need for further comment, I find the paper of acceptable standard for the conference. The authors are encouraged to take note of the suggestions made by the reviewers, especially R2 and R3, when improving the paper for eventual publication.